# Full Sails against Cancer

**DOI:** 10.3390/ijerph192416609

**Published:** 2022-12-10

**Authors:** Angela Mastronuzzi, Alessandra Basso, Giada Del Baldo, Andrea Carai, Andrea De Salvo, Alessandra Bonanni, Italo Ciaralli, Domitilla Elena Secco, Paolo Cornaglia Ferraris

**Affiliations:** 1Department of Pediatric Haematology and Oncology, and Cell and Gene Therapy, Bambino Gesù Children’s Hospital, Istituto di Ricovero e Cura a Carattere Scientifico (IRCCS), 00165 Rome, Italy; 2Neurosurgery Unit, Department of Neurosciences, Bambino Gesù Children’s Hospital, Istituto di Ricovero e Cura a Carattere Scientifico (IRCCS), 00165 Rome, Italy; 3Fondazione Tender to Nave Italia ONLUS, 16128 Genova, Italy

**Keywords:** cancer, adventure therapy, sailing navigation, quality of life, adolescence, healthcare

## Abstract

Background: Cancer is very disruptive in adolescence and hospitalizations interfere with this development stage in becoming independent, developing social relationships, and making plans for the future. A major challenge in the care of adolescents with cancer is being able to enhance their quality of life. The aim of this project is to increase our understanding of how adventure therapy influenced quality of life for adolescents with cancer. Methods: Bambino Gesù Children’s Hospital, in collaboration with the Tender to Nave Italia Foundation (TTNI), has been conducting a unique project, located on a beautiful brigantine of the Italian Navy. Adventure therapy is a form of experiential therapy that consists of various types of adventure, in particular outdoor and sailing activities. Ninety teenagers have been the protagonists of this project to date and filled out two questionnaires about quality of life and self-esteem, before and after the sailing experience. Results: The adventure provides the opportunity for the participants to build interpersonal relationships and develop life skills that they can benefit from in the future experiences. All participants report a significant improvement in their quality of life and self-esteem at the end of this experience. Conclusion: This collaborative adventure project is a great way to learn and practice new behaviors, improve interpersonal skills, heal painful emotions, overcome personal obstacles and challenges, and help the teenagers to resume their developmental path after an onco-hematological diagnosis.

## 1. Introduction

Pediatric onco-hematology is one of the specialties of pediatrics that has seen its greatest development in the last two decades. The better understanding of the pathophysiological mechanisms underlying pediatric neoplasms and, in particular, the genetic and molecular characteristics that distinguish them, has increasingly made it possible to use advanced therapeutic approaches adapted to the patient from the perspective of precision medicine [1].

Faced with better chances of recovery, children who have suffered from cancer often pay the price of the treatment they undergo, showing over time neurological, endocrinological, emotional, and cognitive problems that affect the quality of their life in a greater or less significant way [2]. On the other hand, cancer is always a disruptive disease in the life of the person affected by it; however, when a child is diagnosed with cancer, the whole family becomes ill with them, and when a teenager is diagnosed, there is an additional setback in the physiological process of detachment from parents that leads to independence and to adulthood [3,4].

The main objective towards which the researchers’ efforts are focusing is therefore twofold, namely, to increase the number of children and young people who recover from cancer while simultaneously improving the quality of their life both during hospitalization and follow-up [5].

Therefore, treating the patient’s health, does not simply mean eliminating the disease; the multifactorial dimension of health requires a multi-level approach. It is essential to respect the dignity of the person and to maintain their mental and social well-being, enhancing psychophysical abilities, respecting the environmental, and relational conditions. This is particularly true in developmental age patients and, specifically, in those facing complex diseases that require frequent hospitalizations and follow-up [6,7]. For all these reasons, it is important to involve different professional figures such as pediatric oncologists, nurses, psychologists, pedagogists, and youth workers in order to compose a multidisciplinary team.

In this line of research focused on the quality of life during treatment, the Department of Onco-hematology, Cell Therapy, Gene Therapies and Hemopoietic Transplantation of the Bambino Gesù Pediatric Hospital, with the Tender to Nave Italia Foundation, has been carrying out a project since 2014 for “adventure therapy” called “Full Sail Against Cancer”. The “adventure therapy” is a form of experiential therapy that consists of various types of exciting outdoor activities; the program aims to increase self-esteem and relational skills and at the same time promotes positive change that improves the quality of life and makes the effect of the disease less impactful. For these reasons, adventure therapy has for several years fully entered the “treatment” and “taking care” path of the pediatric and adolescent patients suffering from cancer [8,9].

In this paper, we describe our project and show its effectiveness in improving the quality of life of patients. To do this, we evaluate self-esteem and QOL of participants before and after their experience as we describe below.

## 2. Materials and Methods

### 2.1. “Full Sail against Cancer”: The Name

The name of the project comes from a brainstorming between doctors, psychologists, nurses, and the group of young people. The idea, born from the kids themselves, had the aim of combining the experience of everyday life, that is, the fight against cancer, with an aspect that would refer to the “outdoor” adventure that they were preparing to live and share. They themselves had the expectation that the adventure on Nave Italia could be part of their treatment path, an aid to their daily battle. This is the reason that name has been kept for all the voyages thereafter.

### 2.2. The Objectives

The project, in continuity with the most recent trends of the literature in the psychoeducational field, has set numerous objectives [10].

The goal of the first embarkation was to improve the quality of life in children at different stages of the disease through adventure therapy, and at the same time to increase the awareness of these benefits on the operators.

In subsequent embarkations, the objectives that were set were different and numerous. In particular, great attention was paid to the evaluation of how a challenging adventure without parents could increase self-esteem and the acceptance of one’s own limits and abilities.

A further goal of the project was to strengthen, through taking part in the adventure together, the therapeutic alliance between healthcare professionals and patients and between healthcare professionals themselves.

Finally, the ambitious goal was to evaluate the “educational” impact that children with their stories and sharing these stories have on their peers who are also facing the disease.

### 2.3. The Project

The criteria for selecting participants to the project were: patients between 8–25 years, in treatment or out of therapy, good clinical condition, no severe intellectual disability, no severe cytopenia or other severe toxicity related to treatment, balanced number of males and females, groups of participants suffering from different diagnosis, avoiding the participation of the same patient twice in the activity.

Every year, the protagonists of the project are 15 children and young adults, and a team of professionals from the Department, including a pediatric oncologist, two nurses, a psychologist, and a volunteer. Ninety teenagers have been the protagonists of this project from 2014 to 2019. Detailed data of all participants are described in Table 1. For each edition, new patients were chosen in order to be able to offer this adventure to as many as possible or patients who had already participated in the project but who for physical reasons related to the disease had not been able to fully make the most of the experience (for example, children who at first boarding were carriers of a central venous catheter and therefore could not bathe) or identified in turn as a sort of glue “educators” or binding element for the group of children with respect to how they had lived their first experience on the ship.

Each year a group was chosen as homogeneous as possible by age group and with a clinical condition that allowed, even with limitations, to be able to fully enjoy the experience and that did not need to perform blood chemistry tests. Home therapies were administered by nurses, as well as central venous catheter dressings, antibiotics, and supportive therapies in case of fever.

Patients with all types of onco-hematological diseases were included such as: patients with leukemia undergoing chemotherapy treatment, patients also in the early post-transplantation phase of hematopoietic stem cells, patients with low vision due to gliomas of the optic pathways, and patients neurologically impaired or walking with support due to central nervous system or bone neoplasms. It was also possible to invite mild cytopenic patients with a lower value of white blood cells, red blood cells, and platelets, resulting in therapies but not urgently required within the 5 days of the adventure.

### 2.4. The Nave Italia Methodology Applied to Pediatric Onco-Hematological Patients

On board Nave Italia, each project consists of three phases, before, during, and after navigation: this methodology was designed so that the development processes are really significant for each individual, which requires it to include a certain temporal continuity to develop in the participants’ life skills that they can also use in their everyday lives.

### 2.5. Data Analysis

SPSS software version 26.0 was used for statistical analysis. Mean scores obtained by participants on all scales of the Pediatric Quality of Life (PedsQL) and Test Multidimensionale dell’Autostima (TMA) before and after the intervention were compared using dependent samples *t*-tests. In addition, for each dimension, the mean change observed in the transition between pre- and post-test was indicated to provide indications of the impact of the program at the individual level. In this regard, it was also chosen to report the 95% confidence interval data for the mean of these differences.

## 3. Results

### 3.1. Preparation Activities

As with any trip, the success of an educational project on board is linked to the quality of its preparation ashore. The choice of participants and health professionals to be involved, communication with families, preliminary meetings, and the comparison of individual and group ideas and objectives, represent in the pre-boarding phase an indispensable tool to be developed by declining in detail needs, experiences, plans of intervention, limits, and risks. In consideration of the type of patients, the choice of the participants has always been made close to the departure: this helps the participants not generate expectations that sometimes are impossible to maintain (just think of a recurrence of the disease, a complication that lengthens treatment times, etc.). Part of the preparation activities also involved sharing the ports with the Nave Italia team of embarkation and disembarkation. In fact, the ports affect which types of our patients can go on board the ship. A nearby port favors the parents accompanying their children to and from: this allows for the better overall management of the detachment but also greater flexibility in the identification of patients for example with physical disabilities that would greatly limit travel on other means of transport. Similarly, very long voyages further from the coast limit the involvement of onco-hematological patients who are more at risk of complications (for example, it is less prudent in these cases to accompany cytopenic patients who may have a fever that requires hospitalization).

### 3.2. Navigation

Activities during navigation are organized and provided in relation to the person who has to carry them out. The dimension of the contents and the life on board become the main mediators around which two other aspects of the educational processes evolve: the relationship with others and the collection of materials and ideas for the narration/documentation of one’s path. Relationships with others, especially in the presence of people with interpersonal difficulties, can be based on the conscious construction and use of a network of mediators (emotional, cognitive, practical, concrete) that educators structure to the extent that they know the people involved. With respect to navigation activities, some of these deserve particular note. Bathing in the sea certainly represents a crucial moment for our patients. For many of them, this is the first swim after 1–2 years of therapy. At this moment, there is both enthusiasm and fear of not being able to come out of the water. Added to this is the proportion of young people who, despite wanting to experience the adventure, are carriers of a central venous catheter and therefore cannot go into the water. The sea bath has therefore always been a crucial moment during the ship phase and as such, it is also perceived by the crew who have always sought alternative solutions for those who could not do it or were afraid to get off the ladder. Another crucial moment of the navigation phase is represented by moments of relaxation. In an absolutely unexpected way, we found that all patients have a clear perception of what is happening to them, even those who are sometimes kept in the dark about their situation by their parents. Children and young people are perfectly aware of the pathologies they suffer from and a constant upon boarding was to find them seated at a table grouped by disease or by type of treatment performed. We have heard them discuss the side effects of drugs, the experience of the disease, and above all we heard them expressing their concern towards their parents and their siblings. The opening and closing of the sails take effort but are also a great deal of fun. The possibility of trying something so physical in proportion to one’s possibilities, thanks to the crew, certainly had an important impact on everyone’s perception of themselves and their abilities. The games selected and conducted by the Foundation’s project manager made it possible to bring out many feelings that are sometimes difficult to contain. Karaoke and dance evenings gave our kids a bit of normality in a story that has little to do with “normal”. Receiving their diplomas made them proud of who they are, also thanks to the experience—of the illness and of Nave Italia—they went through.

More specific examples of activities and routes is included in the Appendix A.

### 3.3. Evaluation

The “after”, called the post-boarding phase, focuses on integrating the person’s learning into his/her daily life, through narration and documentation of the course and use of the person in future projects where he will be able to put into practice the skills learned, with the mentoring and guidance of other people. A final important aspect of the methodology is evaluation. Although each project has specific tools, the foundation proposes some common evaluation systems. The evaluation of Nave Italia’s educational projects takes place in three fundamental moments for the development of the training experience described above (before, during, and after boarding).

The measuring instruments can be chosen by psychologists, educators, and health workers of the Bambino Gesù Hospital, in accordance with those of the Foundation, depending on the purpose of the educational project. The first tool is a widely validated tool in Italy for self-esteem and is the self-esteem multidimensional test [11]. Specifically, our patients were administered quality of life tests such as PedsQL v4.0 (pediatric quality of life) [12] with the possibility of comparing different assessments over time as an integral part of the treatment protocols and to verify any differences between the pre- and post-boarding. Both tests documented a clear change in the pre- and post-boarding phases, in self-esteem and self-perception, in a positive sense, which therefore has a direct and positive influence on the quality of life.

In Table 2 and Table 3, we have included data from ninety patients who filled out our questionnaires as an example of how the project has benefits on these two dimensions. As we describe below, our data analysis shows a significant improvement in subscales and total scores of both self-report tools used.

### 3.4. Self-Esteem Driving Change

In the wonderful setting of the brig and thanks to the pre- and post-boarding meetings, the kids had the opportunity to create relationships and develop emotional ties between themselves and with the health workers who accompanied them, thus beginning to perceive that they are not alone in the fight against the disease. The richness of the relationships that it allows to build is one of the peculiar aspects of the sea. Being part of a crew, sharing their thoughts and feelings, fatigue, responsibility, and daily life with others, allows you to enter a new dimension made of collaboration, trust, and cooperation. The days on board are structured in a very precise way, while always maintaining the flexibility to accommodate weather and sea conditions, and various activities follow one another. Some activities are typical of sea life such as those on winds, on nodes, or on the bridge, all conducted by the crew, and others; on the other hand, some activities are specific to the group and conducted in this case by the hospital psychologist, in collaboration with the project manager of the Foundation. These specifically aim to help participants reflect on their experiences and facilitate collaboration and mutual support. The meetings before and after the week on board, on the other hand, have the function of creating a group, preparing for the experience, and then reliving the shared feelings and trying to bring back into everyday life all the skills experienced and learned on board the brig.

What was seen from the beginning of the project was an increase in the self-esteem of the participants, often strongly negatively affected by having such an important pathology and by the changes in their bodies caused by the treatments to which they have been subjected, by long hospitalizations, and for some with important and sometimes destructive surgical interventions. Obviously, difficulties were not lacking, in fact, most of them felt they were unable to bear the rhythms, the rules, and the tasks to be carried out on board together with the crew of the Navy. Others, very shy, found it hard to open up and enter into relationships with people who were even very different from themselves. However, moment by moment, experimenting with physical, emotional, and relational tasks and thus reaching one goal after another, they began to discover some of their abilities, developing a sense of personal empowerment. Furthermore, we noticed that sharing such a strong experience with one’s own care team has also achieved the goal of improving therapeutic compliance and therefore the doctor–patient relationship.

The educational proposal of Nave Italia favors cooperative work, open dialogue between different realities (civil and military world), and inclusive coexistence in an environment as extra-ordinary as that of a wonderful brig: being together, sharing rules and activities of navigation, space, and time, being part of a real crew promotes creative learning and the experiential stake. It gives the opportunity to face fears and limits together and is an excellent educational methodology for learning new behaviors, improving interpersonal skills, overcoming personal obstacles, and healing painful feelings and experiences.

The life on board, the sharing of their space and time, and the specific activities designed by the psychologist and the project manager of the foundation contributed to achieving these objectives.

### 3.5. Some Testimonies of the Adolescents

Before falling ill with acute lymphoblastic leukemia, K.S. practiced swimming at a competitive level. For two years, since the onset of the disease, he had not swum, so he has been waiting with great anxiety for the moment to swim in the sea. After the first few swims, however, K.S. feels his body is tired: he is no longer trained. It is time to accept one’s limits: on board, we improvise a moment, then with the psychologist and the oncologist, we seek a solution and find hope. It was decided to refer K to an intensive muscle physiotherapy program to get back in shape by the time he returns to the pool to train and compete. “I met my limits and tried to overcome them, in nature I tested myself”.

L.M., a girl with Ewing’s sarcoma, has undergone many therapies and treatments, and numerous surgeries to reconstruct her partially deformed face. One night she told her story, not everyone who hears her knows how to name her disease, some because they are too young to understand the word cancer, others because they never wanted to ask either the doctors or their parents. It is the moment of conscience: with the story of L., everyone shaped their fear, regardless of age. Shaping fear also means recognizing something to fight against. “We all cried, we decided to fight together.”

D.D.S. is an 8-year-old boy suffering from optic pathway glioma with consequent loss of vision; when he grows up he wants to be a pirate. For this reason, as soon as he got on board, he asked the captain if he can hoist the pirate flag on Nave Italia, which was immediately granted. After months in the hospital, he does not stop for a moment: he swims, learns to weave lines to form knots, plays with others, and returns to being an 8-year-old. “I love living on this sailboat, my dream is a reality here”.

### 3.6. A.M. Testimony

“I am a pediatric onco-hematologist who has always been involved in an “outdoor” adventure with my patients since my graduation. I have accompanied them on organized trips to amusement parks, on pilgrimages to sanctuaries even outside of Italy, and on recreational days to celebrate events such as the end of school; I organized practical and educational cooking experiences for them; I participate in the creation of the film club and numerous other activities also within the hospital. I consider this attention to be an integral part of the care of my patients. Each of these experiences has its own peculiarities. With regard to the Nave Italia experience, I personally participated in all the onco-hematology embarkations of the Bambino Gesù Children’s Hospital, committing myself personally from the drafting of the project to the selection of participants, to the material organization of travel, to finding funds to carry out the project, to actively participating in Nave activities. I am grateful for the moments I had the privilege of experiencing both because they allowed me to be together with the kids without any filter and because they were the first to put me to the test: physical and emotional fatigue is an important training ground for life. I was able to hear their stories better, not filtered by their parents, and I was pleasantly struck by the depth that characterizes them. I have been a doctor, a friend, a confidant, sometimes a mother, sometimes a sister, and always a traveling companion. The project managers who accompanied us as well as the crew always entered our world gracefully: they allowed themselves to be welcomed by the peculiarity of our group to better guide it in the adventure. Attentive and discreet observers have never lacked their support, sharing their knowledge, point of view, and concrete help with respect to situations of any kind that may have arisen. They are fond of patients and vice versa. They accepted all my fears and helped me overcome them. We created and strengthened the team of health workers: I met aspects of the people who accompanied me that I never could have known in the hospital. Nave Italia is also a team-building experience: it is, therefore, necessary to take great care in the choice of companions which must not be conditioned by mere personal availability but must be aimed at forming a group and strengthening working complicity. Nave Italia for me was being a traveling companion: this is the teaching that I brought back with me to land.”

## 4. Discussion

Tumors are rare diseases in children. In fact, every year in Italy about 1600–1800 children between the ages of 0 and 15 and at least 800–1000 adolescents between the ages of 15 and 19 are diagnosed with a neoplasm [13]. The notable advances in the diagnostic, molecular, and therapeutic fields have made it possible to achieve very high cure rates, which are in excess of 70% [14]. Based on these premises, it is estimated that today, in Italy, one in eight hundred young people is cured of an onco-hematological pathology suffered at a pediatric age.

These successes are the result of international cooperative efforts that have led to a standardization of diagnostic processes and therapeutic protocols, and to a better definition of the risk that the individual pathology has of being resistant to conventional treatments.

The management and treatment of a pediatric neoplasms is very complex and multifaceted, and involves different actors in the various phases. Teamwork is essential to maximize patient outcomes [15]. Furthermore, the role of the pediatric onco-hematologist does not end with the treatment of the disease; the price of the treatments our patients are subjected to is often very high and, in any case, requires specialized and coordinated checks to be carried out over time. The bond that is established between the pediatric oncology center and the patient lasts for a lifetime; we as caregivers continue to be part of the successes and failures of our patients’ personal lives as well. For this reason, unlike other pediatric specialties, it is very complex to structure transition programs.

Pediatric cancers are completely different from their adult counterparts; although they are sometimes referred by the same name, it is now known that the molecular alterations underlying the pathology, epidemiology, and treatment options make these pathologies age-specific [16]. Even within the pediatric age, individual pathologies take on different molecular connotations so as to identify, even in the context of the same family of tumors, typical neoplasms for example of infants or children under one year of age [17,18].

At the same time, adolescence also poses particular problems as it is a borderline age. In fact, adolescents are diagnosed with both cancers typical of the pediatric age (such as leukemia and brain tumors) and tumors typical of adulthood (such as carcinomas and melanomas) [19].

This makes the medical management of these patients even more complex as the team dedicated to adolescents must have not only onco-hematological but pathophysiological skills to treat both the child and the adult. It is therefore essential in this age group to create synergies with onco-hematology centers that follow adults in order to offer the patient the best therapeutic possibilities, which for various reasons is more suited to the adolescent.

Focusing on onco-hematological adolescents, tumors of adolescents and young adults are today referred to as AYA, an acronym for “Adolescent and Young Adults”. Naming this category has resulted in a significant step forward in the knowledge and treatment of very particular patients. The AYAs have been “ghost patients” for years; this is due to the particular borderline in which these neoplasms are placed epidemiologically. This has resulted in a significant breakthrough in the knowledge and treatment of patients.

More than three in four adolescents have a chance of recovery due to a multimodal treatment that includes surgery, chemotherapy, and radiotherapy/radiation therapy. With the same clinical condition (disease and stage), an adolescent is in fact less likely to recover than a child, often simply in relation to a diagnostic delay, the quality of care, and enrollment in clinical protocols.

For the treatment of these patients, it is necessary to have a professional figure who has “all-round” experience of extremely different pathologies and who can guarantee access to all treatment protocols (pediatric and adult).

The onset of the disease in this particularly delicate moment of the growth process causes upheaval in the lives of young people, just when, physiologically, they should be establishing themselves and creating new balances with studies, work, and loved ones [20,21].

A teenager who suffers from cancer is someone who is growing up and the goal of “taking charge”, in addition to healing, is to support them in their natural evolutionary processes, complex and peculiar.

During the treatment and hospitalization phases, the adolescent must seek resources to adapt to a body that, in addition to changing, causes them pain and no longer functions as before, making it necessary to ask for help from the adult world (parents and doctors) from whom they would like to become independent of.

The ways to adapt to the disease are almost infinite and the adaptive or maladaptive quality of these reactions can deeply affect the quality of life of patients. Adaptation is a complex process that occurs gradually, in the balance between worry, anger, the need for normality, and the need for knowledge of the disease and treatment.

Communication and the relationship with the treating physician are crucial steps, to facilitate the processes of adaptation and elaboration of the diagnosis, the starting point of a relationship that will last months and years and on which the therapeutic alliance will be based. The sick adolescent must necessarily be considered a fundamental interlocutor with respect to the medical choices that concern them, seeking the correct balance between the patient’s right (ethical and legal) to be informed and, at the same time, the need to protect them. Every patient of this age must be recognized as an individual with autonomous behaviors and thoughts from those of their parents.

In the psycho-social support of adolescent patients, the role of the school is crucial [22]. It is extremely important not to interrupt the course of study; school continuity provides implicit reassurance for children: there is a future and the treatments are carried out with the intention of healing. You need to inform your schoolmates, help them understand the absence, and plan a conscious welcome when the classmate returns and needs to feel part of a group again.

Treating adolescents means recognizing the complexity of their management and the need to achieve overall support through a multi-specialized team who alongside the oncologist [23], work daily with not only the surgeon, radiotherapist, radiologist, pathologist, endocrinologist, neurologist, but also the psychologist, social worker, educator, entertainer, and spiritual assistant.

Adolescents who were diagnosed in childhood are another group of patients. This type of patient continues to be followed in the pediatric center of reference and begins to experience all the complications resulting from treatment. Adolescents recovered from cancer represent an inexhaustible source of support to their peers but also to younger children who are experiencing the stage of the disease.

## 5. Conclusions

In describing our experience, we showed the validity of the “Full sails against cancer” project and its impact on the quality of life and participants’ self-esteem, reported by our assessment tools. We conducted a data analysis on the self-report tools used by our participants. Our results show a significative positive effect of the project Nave Italia on all evaluated scales of quality of life and self-esteem. For this reason, further investigation studies will be performed in the future to support these data more specifically. Self-report tools may be subject to bias from participants and in the future, maybe it would be useful to provide an additional assessment point further down the road to see if these benefits persist long after landing and returning home. However, we think it would still be found to be useful in better understanding the impact of the project.

The care and management of pediatric patients with onco-hematological pathologies are two fundamental aspects of the recovery process. Healing does not only mean curing, but it also means providing tools to grow and develop one’s talents even in the difficulties that come with having a serious illness. Nave Italia allows the group to stay together, sharing their space and their time in close proximity with nurses and doctors involved in hospital activities, and gives the opportunity to face fears and limitations together. Nature, sea, wind, and adventure, together with the security of military expertise and the rigor of rules to be respected, offer a unique setting to make this multidisciplinary project possible. The Nave Italia method is not only fun, exciting, and stimulating, but a great way to learn and practice new behaviors, improve interpersonal skills, face fears, overcome obstacles and personal challenges, and heal painful memories. We show a significant positive impact of this kind of project on quality of life in our sample. It deserves to be formally included in the therapeutic protocols of advanced scientific research.

## Figures and Tables

**Table 1 ijerph-19-16609-t001:** Demographic data of patients.

Characteristics	N (%)
Male	39 (43%)
Female	51 (57%)
Age mean (range + SD)	14 (13–19 ± 2.64)
Disease:	
− Leukemia	30 (33%)
− Lymphoma	11 (13%)
− Brain tumors	25 (28%)
− Bone sarcoma	15 (17%)
− Neuroblastoma	5 (5%)
− Other	4 (4%)

**Table 2 ijerph-19-16609-t002:** T mean scores in TMA self-report scales (*n* = 90).

TMA T Mean Scores						
	T0	T1	t	*p*	Δx	CI 95%
**Personal**	54	60	−8.07	<0.001	5.63	4.25–7.02
**Skills**	37	45	−9.13	<0.001	7.53	5.89–9.17
**Emotional**	46	50	−7.45	<0.001	4.37	3.20–5.53
**School**	45	46	−3.62	<0.001	1.46	0.66–2.25
**Family**	60	63	−11.28	<0.001	3.41	2.81–4.01
**Body**	40	50	−20.69	<0.001	9.51	8.60–10.42
**Total**	47	52	−22.39	<0.001	5.34	4.87–5.82

**Table 3 ijerph-19-16609-t003:** Mean scores in PEDSQL self-report scales (n = 90).

PEDSQL Self-Report Scales						
	T0	T1	t	*p*	Δx	CI 95%
**Physical** **Health** **mean (range + SD)**	76 (0–100; 22.0)	81 (19–100; 19.3)	−6.40	<0.001	4.49	3.09–5.89
**Emotional** **Functioning** **mean (range + SD)**	76 (40–100; 15.5)	83 (60–100; 14.1)	−8.01	<0.001	6.91	5.20–8.62
**Psychosocial Health** **mean (range + SD)**	85 (65–100; 13.1)	90 (65–100; 12.3)	−6.70	<0.001	5.12	3.60–6.64
**School** **Functioning mean (range + SD)**	77 (45–100; 14.3)	81 (45–100; 15.0)	−10.09	<0.001	4.22	3.39–5.05
**Total** **mean**	79	84	−14.07	<0.001	5.14	4.42–5.87

## Data Availability

Not applicable.

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
