# Peer review of "Full Sails against Cancer"

_ijerph, 2022, doi:10.3390/ijerph192416609_

Round 1
Reviewer 1 Report
The paper is very interesting. I would encourage authors to write a more specific and clear objective in the introduction. And they should provide a descriptive table of the patients, the activities and the pedsQL tests. It would reduce the size of the discussion and focus the discussion on the results. I would also transfer the acknowledgments at the end of the introduction to the end of the text.
Consider the new professional profiles to work on the adventures of these patients.
Consider about the importance of experiences in contact with Nature and the improvement of pedsQL scores, could you present a table?
Author Response
We are pleased that you have considered our work and provided us with very interesting comments that will enhance our paper.
First of all we would like to thank the reviewers for their interesting comments.
The paper is very interesting. I would encourage authors to write a more specific and clear objective in the introduction. And they should provide a descriptive table of the patients, the activities and the pedsQL tests. It would reduce the size of the discussion and focus the discussion on the results. I would also transfer the acknowledgments at the end of the introduction to the end of the text.
Consider the new professional profiles to work on the adventures of these patients.
Consider about the importance of experiences in contact with Nature and the improvement of pedsQL scores, could you present a table?
Thank you for your considerations and comments. We write a more specific and clear objective in introduction section and provide a descriptive table of patients and pedsQL score. The activities carried out during the project are detailed in supplementary section.
Moreover, we reduced the size of discussion and transfer the testimony of patients and tutors in results section.
Finally, we transfer the acknowledgments at the end of the text, as you suggested.

Reviewer 2 Report
This manuscript provides an overview of a very interesting program for young cancer patients. As an introduction to the program, it serves very well. As a report on the impact of the program, and its potential for use elsewhere, it has a number of limitations:
1) It would have been helpful had more detail been provided about processes used to identify potential participants for the yearly program.
2) It would have been of interest to have been provided examples of a typical itinerary of the program (perhaps a brochure from a recent program as an appendix in the supplementary materials?).
3) Given that this program appears to have received appropriate approvals, it would be very interesting to have seen summaries of outcome measures from the tools used to gauge quality of life, and so forth, as part of an assessment of the impact on the individual participants. This would provide a crucial link between the results of the program that could be linked to a conclusion section that would have greater impact.
Author Response
We are pleased that you have considered our work and provided us with very interesting comments that will enhance our paper.
First of all we would like to thank the reviewers for their interesting comments.
1) It would have been helpful had more detail been provided about processes used to identify potential participants for the yearly program.
Thank you for your comment, we added more details about processes used to identify participants.
2) It would have been of interest to have been provided examples of a typical itinerary of the program (perhaps a brochure from a recent program as an appendix in the supplementary materials?).
Thank you for your advice, we added a typical itinerary of the program and activities in the supplementary section.
3) Given that this program appears to have received appropriate approvals, it would be very interesting to have seen summaries of outcome measures from the tools used to gauge quality of life, and so forth, as part of an assessment of the impact on the individual participants. This would provide a crucial link between the results of the program that could be linked to a conclusion section that would have greater impact.
Thank you for your suggestion, we added a table with quality of life score for the population of participants to the project.

Round 2
Reviewer 1 Report
You have made an effort to improve the work.We hope that in the future they will be able to present the progress achieved. I suggest that you include some statistic (p) in table 3 and highlight the significant results in the summary.
Author Response
Thank you for your new comment.
We added p-value of test results in the table and we discussed it also in the summary.
Reviewer 2 Report
This revised manuscript contains a number of important modifications that greatly strengthen its likely impact. It now provides more information about participant selection and more information about the programs that are implemented for the patients. It also provides some summary data that provides insights into the potential impact of the program.
If two additional items were included, the quantitative summaries would be enhanced even further:
1) It would be informative to include the number of responses included in each of the summary statistics that are presented in Tables 2 and 3.
2) As there is a lot of person-to-person variability in the scores reported in Tables 2 and 3, it would be very helpful if summaries of the post minus pre change scores (e.g. means and 95% confidence intervals of the changes observed within each participant) were provided. This would give a clearer picture of the likely impact of the program for an individual patient.
3) The demographic data that are presented in Table 1 is not reflective of the stated 16 children per year. It would be informative if counts of the number of participants per year were added to this Table.
Author Response
Dear reviewer, thank you for your suggestions.
1) It would be informative to include the number of responses included in each of the summary statistics that are presented in Tables 2 and 3.
Thank you for this suggestion. All 90 patients answered the test questions they were subjected to. We specified the number in the table caption.
2) As there is a lot of person-to-person variability in the scores reported in Tables 2 and 3, it would be very helpful if summaries of the post minus pre change scores (e.g. means and 95% confidence intervals of the changes observed within each participant) were provided. This would give a clearer picture of the likely impact of the program for an individual patient.
Thank you for your advice, we added 95% confidence intervals, p-value and misure deviation (Δx) for each item. The mean value with range and sd was already described in the table.
3) The demographic data that are presented in Table 1 is not reflective of the stated 16 children per year. It would be informative if counts of the number of participants per year were added to this Table.
From 2014, the project involved 90 participants. Considering COVID emergency, Nave Italia program was interuppted in 2020 and 2021. Data of QoL and self esteem for 2022 edition are ongoing and we decided do not include these patients in the description.
Ultimately, from 2014 to 2019, there were 90 participants. We specified the year range in the manuscrupit: "Ninety teenagers have been the protagonists of this project from 2014 to 2019".
Globally we have data from 90 patients in total 6 editions of the project. There are 15 participants per year, not 16 (typo that we have corrected in the manuscript).